# NEAT1 Long Isoform Is Highly Expressed in Chronic Lymphocytic Leukemia Irrespectively of Cytogenetic Groups or Clinical Outcome

**DOI:** 10.3390/ncrna6010011

**Published:** 2020-03-13

**Authors:** Domenica Ronchetti, Vanessa Favasuli, Paola Monti, Giovanna Cutrona, Sonia Fabris, Ilaria Silvestris, Luca Agnelli, Monica Colombo, Paola Menichini, Serena Matis, Massimo Gentile, Ramil Nurtdinov, Roderic Guigó, Luca Baldini, Gilberto Fronza, Manlio Ferrarini, Fortunato Morabito, Antonino Neri, Elisa Taiana

**Affiliations:** 1Department of Oncology and Hemato-oncology, University of Milan, 20122 Milan, Italy; domenica.ronchetti@unimi.it (D.R.); vanessa.favasuli@unimi.it (V.F.); ilaria.silvestris@unimi.it (I.S.); luca.agnelli@unimi.it (L.A.); luca.baldini@unimi.it (L.B.); elisa.taiana@unimi.it (E.T.); 2Hematology, Fondazione Cà Granda IRCCS Policlinico, 20122 Milan, Italy; 3Mutagenesis and Cancer Prevention Unit, IRCCS Ospedale Policlinico San Martino, 16132 Genova, Italy; paola.monti@hsanmartino.it (P.M.); paola.menichini@hsanmartino.it (P.M.); gilberto.fronza@hsanmartino.it (G.F.); 4Molecular Pathology Unit, IRCCS Ospedale Policlinico San Martino, 16132 Genova, Italy; giovanna.cutrona@gmail.com (G.C.); monica.colombo@hsanmartino.it (M.C.); serena.matis@hsanmartino.it (S.M.); 5Hematology Unit, Department of Onco-Hematology A.O. of Cosenza, 87100 Cosenza, Italy; massim.gentile@tiscali.it; 6Centre for Genomic Regulation (CRG), The Barcelona Institute of Science and Technology, Dr. Aiguader 88, 08003 Barcelona, Catalonia, Spain; ramil.nurtdinov@crg.eu (R.N.); roderic.guigo@crg.eu (R.G.); 7Department of Experimental Medicine, University of Genoa, 16126 Genoa, Italy; ferrarini.manlio@gmail.com; 8Unità di Ricerca Biotecnologica, Azienda Sanitaria Provinciale di Cosenza, 87051 Aprigliano (CS), Italy; f.morabito53@gmail.com; 9Department of Hematology and Bone Marrow Transplant Unit, Augusta Victoria Hospital, 97300 Jerusalem, Israel

**Keywords:** NEAT1, Chronic Lymphocytic Leukemia, lncRNA

## Abstract

The biological role and therapeutic potential of long non-coding RNAs (lncRNAs) in chronic lymphocytic leukemia (CLL) are still open questions. Herein, we investigated the significance of the lncRNA NEAT1 in CLL. We examined NEAT1 expression in 310 newly diagnosed Binet A patients, in normal CD19+ B-cells, and other types of B-cell malignancies. Although global NEAT1 expression level was not statistically different in CLL cells compared to normal B cells, the median ratio of NEAT1_2 long isoform and global NEAT1 expression in CLL samples was significantly higher than in other groups. NEAT1_2 was more expressed in patients carrying mutated *IGHV* genes. Concerning cytogenetic aberrations, NEAT1_2 expression in CLL with trisomy 12 was lower with respect to patients without alterations. Although global NEAT1 expression appeared not to be associated with clinical outcome, patients with the lowest NEAT1_2 expression displayed the shortest time to first treatment; however, a multivariate regression analysis showed that the NEAT1_2 risk model was not independent from other known prognostic factors, particularly the IGHV mutational status. Overall, our data prompt future studies to investigate whether the increased amount of the long NEAT1_2 isoform detected in CLL cells may have a specific role in the pathology of the disease.

Chronic lymphocytic leukemia (CLL) has a highly heterogeneous clinical course, ranging from an indolent behavior to an aggressive disease that needs prompt treatment in almost 30% of cases. These differences have been associated with a number of markers of the leukemic cells, including chromosomal aberrations, mutational status of the Immunoglobulin heavy chain variable region genes (*IGHV*), TP53 inactivation, CD38 and ZAP-70 expression [1]. However, despite the availability of these markers, the disease course remains somewhat unpredictable.

In the recent years, attention has been focused on long non-coding RNA (lncRNA), which are involved in many biological processes, such as transcriptional gene regulation, cell development and differentiation. Deregulation of lncRNAs has been demonstrated to be connected with tumor formation, progression and metastasis in many types of cancers, including hematological malignancies, although the information on a potential pathogenetic role in CLL is rather limited [2,3,4,5]. 

In this study, we focused on nuclear paraspeckle assembly transcript 1 (NEAT1), a well-known lncRNA located on chromosome 11q13. NEAT1 is transcribed in two different isoforms: a canonically polyadenylated short transcript of 3.7 kb (NEAT1_1), and a longer non-polyadenylated transcript (NEAT1_2) of about 23 kb that includes entirely the short NEAT1_1 form. The two isoforms share a common promoter but have an alternative transcription termination site. NEAT1_2 is an indispensable structural component of paraspeckles (PSs), which are membraneless compartments of the nucleus [6]. Although their function is not fully defined, PSs are involved in stress response and influence gene expression by regulating both transcription and pre-mRNA splicing events and by holding nuclear mRNA for editing [7]. NEAT1_2 could indirectly control these events by modulating the functions of PSs upon exposure to specific stresses [8]. Concerning NEAT1_1, even if it represents the most abundant isoform found in all samples, its biological role has to be fully elucidated. Recent data strongly suggested that it could be nonfunctional [8] leading to the hypothesis that NEAT1_1 keeps the transcription of the NEAT1 locus active, guaranteeing a rapid switch to NEAT1_2 production in response to stress. 

NEAT1 deregulation has been reported in many types of solid tumors, where it is often associated with a poor prognosis, and in hematological malignancies, where it appears to affect different biological processes. Specifically, the aberrant expression of PML-RARα activity is correlated with NEAT1 downregulation in acute promyelocytic leukemia, suggesting that it may contribute to the impairment of myeloid differentiation [9]. We recently reported that the expression of NEAT1 in multiple myeloma (MM) is well above the normal controls, although this deregulation does not appear to correlate with prognosis. However, the putative NEAT1 involvement in different mechanisms of cellular stress response, such as the Unfolded Protein Response (UPR) and TP53 pathways, makes it a confident candidate for a potential targeted therapy in the disease [10,11]. Moreover, the high NEAT1_1 levels observed in MM suggest the possibility of NEAT1_1 involvement in PSs unrelated functions. 

Information on NEAT1 expression and its possible deregulation in CLL is still lacking. Recently, Blume et al reported that NEAT1 expression can be induced during DNA damage responses in CLL cases with an intact TP53 function [12]. To gain further information on this issue, we investigated NEAT1 expression in 310 newly diagnosed Binet A patients prospectively enrolled in an observational multicenter study (clinicaltrial.gov #NCT00917540 from January 2007 to May 2011) (Table 1) [13]. The National Cancer Institute (NCI)-sponsored Working Group guidelines were followed for diagnosis and staging [1]. Eighty four of these 310 cases, who had less than 5.0 × 10^6^ monoclonal B lymphocytes/L in the blood, were reclassified and diagnosed as monoclonal B-lymphocytosis (MBL) in accordance with the more recent International Workshop on Chronic Lymphocytic Leukemia (IWCLL) diagnostic criteria [1]. Median follow-up time was 76 months (range, 1–130 months). Highly enriched CD19+ CLL cells were characterized for IGHV mutational status and cytogenetic alterations, including deletion of 13q (del13), 11q (del11), and 17p (del17) and trisomy of chromosome 12 (12+), as previously reported [14]. *NOTCH1* mutations were also investigated as described [15].

In addition, we evaluated NEAT1 expression also in other types of hematological tumors, including B-cell acute lymphoid leukemia (ALL), acute and chronic myeloid leukemia (AML and CML), MM, B cell-lymphoma cell lines, and different types of normal B-cell populations, i.e., 27 samples including normal peripheral blood B-cells (pBC) and naïve and memory B cells purified from spleen or tonsils as specified elsewhere [14]. All the statistical tests were performed using appropriate R functions setting *p*-value < 0.01 as cutoff for significance.

For NEAT1 determination a quantitative real-time PCR (qRT-PCR) approach was used that was capable of discriminating the NEAT1 (NEAT1_1 and NEAT1_2) global expression from that of the NEAT1_2 isoform [11]. Global NEAT1 expression was also confirmed by RNA FISH (Appendix A). 

The NEAT1 expression levels in CLL cells are shown in Figure 1A also in comparison with those found in normal B-cell populations and in the malignant cells from the other hematological tumors. NEAT1 and NEAT1_2 expression levels were not statistically different in CLL cells compared to normal B cells. No differences could be detected even when CLL cells were compared separately with either normal naïve or memory B cells, which are considered closer to CLL cells (Appendix A) [14,16]. CLL cells expressed significantly more NEAT1 than those of the other hematological neoplasias with the exception of MM cells, which are known to express high levels of this lncRNA (Figure 1a, left panel) [17]. Next, we verified the correlation between the two NEAT1 isoforms in all the populations analyzed and, with the exception of the small group of B-lymphoma cell lines, we found that NEAT1_2 expression levels positively correlated with those of NEAT1_1 (Figure 1b). Interestingly, CLL cells expressed the highest amount of NEAT1_2 isoform compared to the other cell types (Figure 1a, right panel). In particular, the median ratio of NEAT1_2 and NEAT1 expression in CLL samples (40%) was significantly higher than in the other groups (range 5–21%, Figure 1c). 

Although, on the whole, our CLL series had a median NEAT1 expression similar to that of normal B cells, a proportion of samples showed high expression levels of global NEAT1 (Figure 1a, left panel) or NEAT1_2 (Figure 1a, right panel) long isoform. Prompted by such findings, we investigated whether differences in NEAT1 expression could correlate with other characteristics, which usually define different CLL prognostic groups. The global NEAT1 expression was comparable in CLL and in MBL cases, and there was no significant difference in IGHV-mutated (M) or -unmutated (UM) cases or between cases with different cytogenetic alterations (Figure 2a, upper panel; and Appendix A). In contrast, the expression of the long NEAT1_2 isoform was significantly different in CLL subgroups stratified according to prognostic markers (Figure 2a). Specifically, NEAT1_2 was more expressed in the IGHV-M than in the IGHV-UM cases and in CLL cases without cytogenetic aberrations or with the 13q deletion, whereas it was significantly lower in patients with 12+ (Figure 2a, lower panel). No difference in NEAT1_2 expression was observed in the groups with or without *NOTCH1* mutations (Figure 2a, upper panel) [15]. As for global NEAT1, NEAT1_1 isoform did not show any significantly differential expression in all the subgroups investigated (data not shown).

In addition, we investigated the possible association between NEAT1 and *TP53* expression, based on data reporting NEAT1 as an effector of p53 protein, likely playing an important role in suppressing transformation in response to stress signals [18]. To do this, we focused our attention on cases harboring del17p in our series. Although the analysis has been limited only to the four available patients with del17p, our results showed that neither global NEAT1 nor NEAT1_2 expression levels in CLL were significantly lower than the ones detected in patients without del17p (*n* = 299; NEAT1 with del17p: 1.067 ± 1.013 *vs* without del17p 1.628 ± 1.229, *p* = 0.25; NEAT1_2 with del17p: 2.679 ± 0.557 vs. without del17p 3.154 ± 1.576, *p* = 0.34). To better characterize the *TP53* status in these 4 patients, we sequenced the gene and found *TP53* mutations in all samples, with a Variant Allele Frequency (VAF) higher than 95% in three cases (Appendix A). Therefore, in these 4 patients, *TP53* gene appears to be completely disrupted. Overall, these results are in keeping with data by Blume et al. [12], showing that basal NEAT1 expression level is quite similar in CLL patients with a wild-type *TP53* status or in those carrying *TP53* alteration (i.e., mutation and/or deletion). Next, we evaluated whether mutated TP53 proteins found in our 4 patients were capable to activate NEAT1 transcription, by exploiting a yeast-based P53 functional assay (Appendix B) [19,20,21]. Firstly, a new reporter yeast strain (yLFM-NEAT1) in which the p53 response element (RE) from the NEAT1 target gene (5’-GAGCAAGCCTGGGCTTGCCA-3’) [18] controls the expression of the LUC1 reporter gene, was constructed. Whereas wild-type P53 confirmed the ability to activate transcription in yLFM-NEAT1 (Appendix A), all four P53 mutants encoded by the corresponding *TP53* mutations failed to activate transcription in yLFM-NEAT1 (Appendix A). Therefore, it is possible to speculate that a significantly lower NEAT1 expression level in patients harbouring a completely inactive *TP53* mutation with respect to patients without *TP53* alterations is detectable only upon P53 induction by stress, as also suggested by Blume et al. [12].

Lastly, we correlated NEAT1 expression levels with time to first treatment (TTFT) as clinical outcome. To this end, patients were subdivided into sextiles based on global NEAT1 or NEAT1_2 specific expression by leukemic cells. We found that NEAT1 expression was not associated with prognosis. Patients with the lowest NEAT1_2 expression (1st sextile) displayed the shortest TTFT if compared with all the other samples (Figure 2b); however, a multivariate regression analysis showed that the NEAT1_2 risk model was not independent from other known prognostic factors, particularly the IGHV mutational status (Figure 2c).

In conclusion, our study, performed in a large and well-characterized cohort of early stage Binet A CLL patients, has provided evidence that NEAT1 expression levels are quite heterogeneous irrespectively of cytogenetic groups or clinical outcome. Based on these findings and the suggestion that the two NEAT1 transcripts may have different biological roles [8,22], it would be of interest to investigate whether the increased amount of the long NEAT1_2 isoform detected in CLL cells may have a specific role in the pathology.

## Figures and Tables

**Figure 1 ncrna-06-00011-f001:**
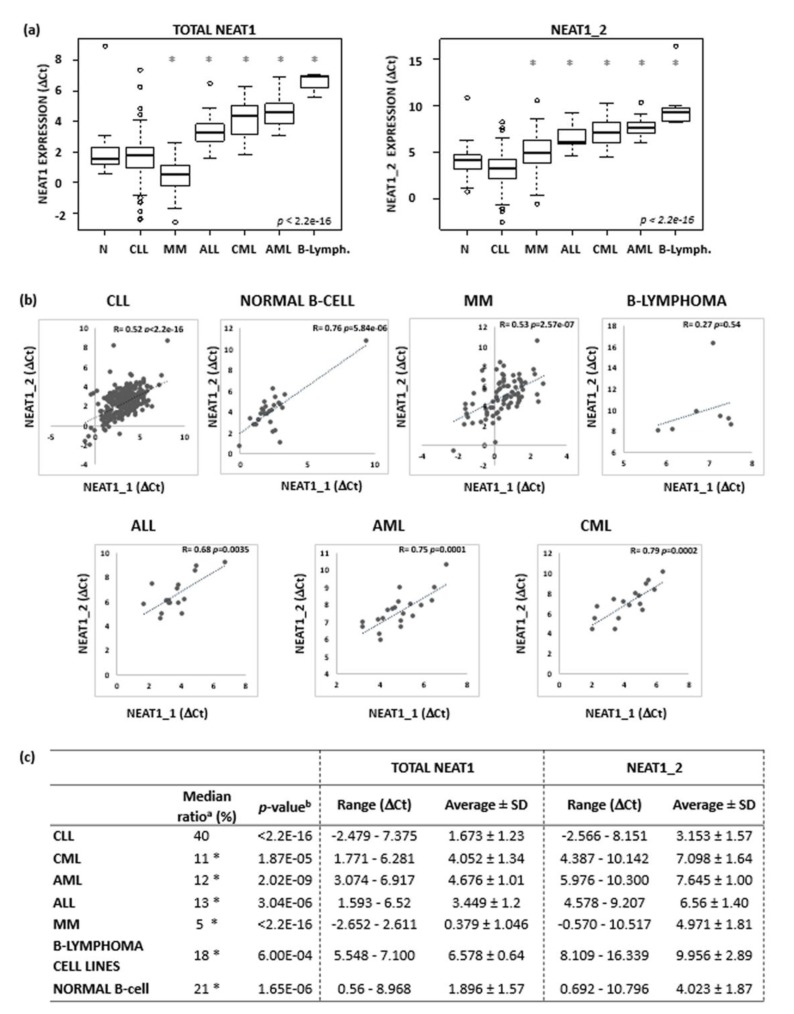
NEAT1 and NEAT1_2 expression levels in normal B-cells and in B-cell malignancies. (**a**) Boxplots of NEAT1 and NEAT1_2 expression levels evaluated by qRT-PCR in 27 normal B-cells, 310 CLL, 82 MM, 16 ALL, 16 CML, and 20 AML samples, and 7 B-lymphoma cell lines (OCILY7, MAVER1, JEKO, MINO, SULTAN, P3HR1, and NAMALWA). Expression data are reported as Ct referred to *GAPDH* housekeeping gene. Significant differences versus CLL group are indicated by an asterisk (Benjamini-Hochberg adjusted Dunn’s test, *p* < 0.01). (**b**) Pearson’s correlation on NEAT1_1 (x-axis) and NEAT1_2 (y-axis) expression level expressed as Ct. NEAT1_1 expression values are inferred as described in Appendix C. Correlation coefficient and p-values are reported in each plot. (**c**) Ratios of NEAT1_2 and total NEAT1 expression level; range, average and standard deviation are reported for each group. ^a^ Median value of the ratios of NEAT1_2 and NEAT1 expression level evaluated for each sample in the specified subgroups. Significant differences versus CLL group are indicated by an asterisk (Benjamini-Hochberg adjusted Dunn’s test, *p* < 0.01). ^b^ Significant differences of NEAT1 and NEAT1_2 expression levels (Wilcoxon test).

**Figure 2 ncrna-06-00011-f002:**
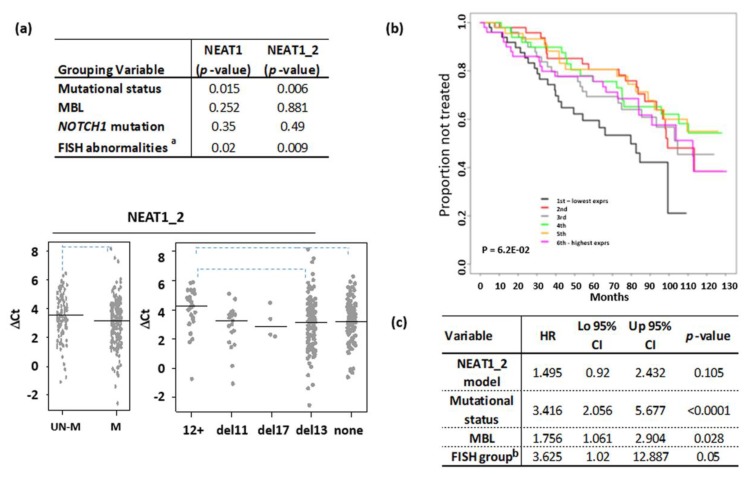
NEAT1 expression level in CLL. (**a**) Wilcoxon test results comparing CLL subgroup defined by the indicated parameter; *p*-value < 0.01 was considered significant (^a^Kruskal-Wallis test). Below, stripchart of NEAT1_2 expression in CLL subgroups defined by mutational IGVH status or the presence of the main chromosomal abnormalities detected by FISH; none = absence of FISH abnormalities (dashed line for *p* < 0.01, Dunn’s test). (**b**) Kaplan–Meier estimated curves of the six groups defined by NEAT1_2 expression levels. (**c**) Multivariate analysis comparing the NEAT1_2 risk model with prognostic variables or with MBL status in CLL series. ^b^ del17 or del11 CLL vs. others.

**Table 1 ncrna-06-00011-t001:** Features of 310 CLL samples.

Parameter	Test Cohort
Number of patients	310
Median age, years (range)	61 (18-71)
Male gender (%)	182 (59)
MBL (%)	84 (27)
IGHV unmutated (%; n.d.)	101 (34; 15)
absence of FISH abnormalities ^a^ (n.d.)	114 (7)
del13 ^b^ (n.d.)	146 (8)
12+ (n.d.)	32 (7)
del11q (n.d.)	19 (7)
del17p (n.d.)	4 (7)
NOTCH1 mutation (n.d.)	46 (1)

^a^ Samples with none of classical cytogenetic aberrations detected by FISH; ^b^ Biallelic 13q deletion was present in 18 samples.

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
