# Peer review of "NEAT1 Long Isoform Is Highly Expressed in Chronic Lymphocytic Leukemia Irrespectively of Cytogenetic Groups or Clinical Outcome"

_ncrna, 2020, doi:10.3390/ncrna6010011_

Round 1

Reviewer 1 Report

The manuscript is well written and presented. Authors mainly show the ratio of NEAT1_2 long isoform and NEAT1 global expression is significantly higher in CLL samples compared to other samples. Also, they state low NEAT1_2 seemed to be associated with shorter survival, but that is not independent from other risk factors (biological and clinical) in CLL. Low correlation coefficient listed in figure 1 (CLL) also suggests that the ratio is very heterogeneous.

Minor.

In figure 1, it would have been better to show NEAT1 total expression and NEAT1_2 isoform expression as a scatter plot. They are not independent, ad showing them in boxplots separately is misleading.

Supplementary figure 1 is actually presenting the main point of the paper, the ratio between NEAT1_2 isofrom and NEAT1 total expression. I think it should be moved to main figures.

Author Response

Referee #1 (Comments to the Author):

- In figure 1, it would have been better to show NEAT1 total expression and NEAT1_2 isoform expression as a scatter plot. They are not independent, ad showing them in boxplots separately is misleading.

As suggested by the Reviewer, we have modified Figure 1 accordingly. In details, in the revised Figure 1 version the panel 1b shows the correlation between the two NEAT1 isoforms as a scatter plot.

Additionally, to address the requests by Referee #2, Figure 1 has been further improved, i.e.:

  • Pearson’s correlation has been performed between NEAT1_1 (values inferred as described in the Appendix B of the revised version) and NEAT1_2.
  • All RT-PCR expression data have been adjusted for PCR primers efficiency, evaluated as described in the Appendix B of the revised version.
  • RT-PCR adjusted values have been plotted in log2 scale.

- Supplementary figure 1 is actually presenting the main point of the paper, the ratio between NEAT1_2 isoform and NEAT1 total expression. I think it should be moved to main figures.

Since the ratio between NEAT1_2 isoform and NEAT1 total expression is reported in “Supplemental Table 1” whereas “Supplementary figure 1” shows NEAT1 detection by RNA-FISH, we assumed that the Reviewer referred to the Supplemental Table 1. Accordingly to Reviewer’s suggestion, these data have been included in the main text of the revised version as Figure 1c.

Additionally, following the request of Referee #3, these data were implemented with range, average and standard deviation for each group.

Reviewer 2 Report

Brief summary:

Ronchetti and collaborators analyzed the expression of the long non-coding RNA (lncRNA) NEAT1 in chronic lymphocytic leukemia (CLL) and various types of leukemias and relevant cell types to understand if the characteristic expression of the lncRNA could be used as a risk factor. They found that differential expression of the long isoform of NEAT1 (NEAT1_2) was associated with other risk factors.

Broad Comments:

The authors explore the expression of NEAT1 in a wide variety of leukemias for a well-characterized cohort of patients, which makes the study very relevant from the clinical standpoint. The authors also made a detailed molecular characterization of the expression of lncRNAs, which represent an underexplored area of research that could yield new insights into different aspects of CLL pathology. The main technical challenge was the dissection of the two different NEAT isoforms, given their overlap, that precluded the authors of an independent quantification of the short isoform. Apart from measuring expression levels, the authors did not explain the characteristic molecular function of the lncRNA in the context of CLL. However, the main concern is that since NEAT1_2 expression is associated with other risk factors, its value as a marker is still unknown.

Specific comments:

Major comments:

The work is based on qPCR analysis of lncRNAs, where the amplification of two different probe sets are directly compared over a wide range of transcript's amount. The authors should provide the amplification efficiency of each amplicon. If 100% amplification was assumed, the authors should make calibration curves for the two amplicons (and housekeeping gene), define the values empirically, and recalculate the expression values for all experiments. Also, the GAPDH primers used should be indicated.

Figures 1a and 1b show the same data plotted in different ways. Please plot data only once and add a table for significance if necessary.

The authors claim that there is always a positive correlation between NEAT1 and NEAT1_2. In the case of CLL, the correlation is rather weak and is also confounded by the fact the NEAT1_2 is counted within NEAT1. The authors should ask whether there is a correlation between NEAT1_1 and NEAT1_2, by substracting NEAT1_2 expression to NEAT1.

The authors show that there is not a significant difference of NEAT1 expression between mutational status (although p Val is < 0.05). The authors should repeat this analysis using the inferred NEAT1_1 expression, as indicated in the previous point, to avoid NEAT1_2 biases.

The authors claim that patients with the lowest NEAT1_2 displayed the shortest TTFT with a p-value of 0.038. Also, the authors decide on a p-value of 0.01 for significance. I find these statements at odds and need to be clarified.

The authors conclude that the increased amount of NEAT1_2 may have a specific role in the pathology. However, the patients with the lowest expression of NEAT1_2 have a potentially worse prognosis (lowest proportion of non-treated). This relationship should be further discussed. Could it be possible that an increased amount of NEAT1_2 may have a protective role in CCL?

Minor comments:

Line 79: include the (MBL) acronym.

Figure 1: give axis a label and plot in log2 scale. The same is true for other qPCR plots.

Supplementary Figure 4: Show points for individual measurements.

Author Response

- The work is based on qPCR analysis of lncRNAs, where the amplification of two different probe sets are directly compared over a wide range of transcript's amount. The authors should provide the amplification efficiency of each amplicon. If 100% amplification was assumed, the authors should make calibration curves for the two amplicons (and housekeeping gene), define the values empirically, and recalculate the expression values for all experiments. Also, the GAPDH primers used should be indicated.

The Reviewer correctly highlighted that the amplification efficiency of each amplicon should be considered. We apologize for having omitted to describe in details the qRT-PCR procedure in the Methods section. Indeed, we had evaluated RT-PCR primers efficiency and found 100% for GAPDH, 106% for NEAT1, and 120% for NEAT1_2, which are in the range 80%-120%, commonly accepted to perform comparisons. However, in order to be as precise as possible, we adjusted all PCR values for primers efficiency and repeated the analyses. Thus, all data reported in the revised version have been amended for amplification efficiency. Information concerning primers and evaluation of efficiency is given in the Appendix B of the revised version of the manuscript.

- Figures 1a and 1b show the same data plotted in different ways. Please plot data only once and add a table for significance if necessary.

As requested by the Reviewer, Figure 1 has been accordingly modified. In details, to sum up requests by Reviewer#1 and #2, panel 1b now reports the correlation between the two NEAT1 isoforms as a scatter plot.

Furthermore, Figure 1 has been improved, i.e.:

  • Pearson’s correlation has been performed between NEAT1_1 (inferred as described in the Appendix B of the revised version) and NEAT1_2.
  • All PCR expression data have been adjusted for PCR primers efficiency, as described in the Appendix B of the revised version.
  • Adjusted values have been plotted in log2 scale.

Additionally, as requested by Reviewer#1, Figure 1 of the revised version includes panel 1c reporting the ratio between NEAT1_2 isoform and NEAT1 total expression.

Finally, following the request of Referee #3, these data were implemented with range, average and standard deviation for each group.

- The authors claim that there is always a positive correlation between NEAT1 and NEAT1_2. In the case of CLL, the correlation is rather weak and is also confounded by the fact the NEAT1_2 is counted within NEAT1. The authors should ask whether there is a correlation between NEAT1_1 and NEAT1_2, by substracting NEAT1_2 expression to NEAT1.

Following the Reviewer’s suggestion, we performed the correlation analyses between NEAT1_2 and inferred values for NEAT1_1. These results have been reported in Figure 1b of the revised version.

- The authors show that there is not a significant difference of NEAT1 expression between mutational status (although p Val is < 0.05). The authors should repeat this analysis using the inferred NEAT1_1 expression, as indicated in the previous point, to avoid NEAT1_2 biases.

As requested by the Reviewer, we performed all the comparisons also with inferred values for NEAT1_1 short isoform, and we did not find any significantly different expression among subgroups, including mutational status (p=0.08).

For completeness, we added this comment in the text lines 145-147.

- The authors claim that patients with the lowest NEAT1_2 displayed the shortest TTFT with a p-value of 0.038. Also, the authors decide on a p-value of 0.01 for significance. I find these statements at odds and need to be clarified.

Indeed, we chose 1% as a cut-off for significance. We apologize in that we have not clearly stated in the original version of the manuscript that p=0.038 should not be considered significant. The threshold for significance in the revised version has been retained at p=0.01; based on Reviewer#3’s criticism, we have included new analysis on TTFT and evaluated its significance accordingly. 

- The authors conclude that the increased amount of NEAT1_2 may have a specific role in the pathology. However, the patients with the lowest expression of NEAT1_2 have a potentially worse prognosis (lowest proportion of non-treated). This relationship should be further discussed. Could it be possible that an increased amount of NEAT1_2 may have a protective role in CCL?

Based on previous point (and as we discuss in response to Reviewer#3), given in any case the lack of significance in the association between NEAT1_2 expression and prognosis, for completeness we analyzed the possible correlation between NEAT1 expression and TTFT either as a continuous variable or stratifying expression into n (2 up to 6) groups. Neither NEAT1 continuous expression nor any of the stratification correlated with outcome. The stratification into sextiles (close to the 15% suggested) was the only one that might indicate a trend of inferior TTFT associated with the lowest-expression sextile (if compared with all the other samples). However, such association was not absolutely retained in a multivariate model. Ultimately, we would not draw any daring hypothesis on the role of NEAT1_2 in CLL.

 Minor comments:

  • Line 79: include the (MBL) acronym.

Done

  • Figure 1: give axis a label and plot in log2 scale. The same is true for other qPCR plots.

Done

  • Supplementary Figure 4: Show points for individual measurements.

Done

Reviewer 3 Report

In the paper by Rochetti et al., the expression levels of the long non-coding RNA NEAT1_1 and NEAT1_2 are probed in different leukemic tumors, particularly CLL. The authors find no significant differences in total expression of NEAT1 (NEAT1_1 + NEAT1_2) or NEAT1_2 alone, but when looking at different liquid tumor subtypes, they find an increase of NEAT1_2 in IGHV mutated, cytogenetically normal and 13q deleted patients but not 12+. Although the 25% lowest NEAT1_2 expressors received treatment significantly faster than the rest of the patients (top 75%), this observation was not independent of other factors; in particular IGHV mutational status (i.e. HR for NEAT1_2 model: 1 (no difference); HR for IGHV mutation status: 3.5). In addition, mutated p53 in the del17 patients does not activate NEAT1 transcription in a yeast-based assay. The authors conclude that NEAT1_2 expression can subdivide CLL patients into subgroups which warrant further investigation for its role in CLL pathology.

Main feel for the paper:

Overall, although my impression is that the data and paper are well-intended, I found this paper hard to read and understand; I did not think the methods were reported properly, if at all, and the main conclusions were vague and/or overstated. Please find my more specific comments below:

Major issues:

Experimental methods, primers used, including efficiencies and ratio calculations are not reported. Choices of statistical tests are not justified (e.g. Rejection of Normal Distribution). See also below comment on pairwise comparisons. Conclusions are either vague or largely overstated; i.e. in the abstract the authors claim that based on the data NEAT1 deserves further investigation on its possible specific role in CLL pathology, but most of the data presented in the manuscript shows no difference in expression or survival for CLL, and no independence to factors already known to be prognostic for patient outcomes. Similarly, the paper does not provide sufficient evidence to state that NEAT1_2 expression defines CLL subgroups. It is unclear why the authors chose to construct Kaplan Meiers of the top 75% of NEAT1 expressors with the bottom 25%. Could the authors construct the model with e.g. top 15% and bottom 15%? Does the separation occur in that case too, which would indicate it is because of NEAT1 expression, and not something else?

Minor issues:

Line 62: cite appropriate references. Intro: introduce genetic architecture of NEAT1 gene (i.e. different isoforms and their proposed functions). Line 71: please cite appropriate references for NEAT1 – p53 link. Line 79 : indicate abbreviation for MBL All qPCR results: using the 2^-dCT value on the Y-axis is less straightforward to interpret than using expression fold changes. Line 108 – 110: Please report range, averages, and standard deviations for all samples. Fig 1A: I am not sure about the author’s choice to compare their expression data in other tumors to those in CLL – it would make more sense to me to compare the data to the normal tissue counterparts if the authors would like to make a point about NEAT1’s importance for cancer. Fig 2A: see comment above, the pairwise comparisons do not make sense to me – what is the biological rationale for comparing with 12+ vs the others? Is this the right control group? Please comment on the consistent higher levels of total NEAT1 versus NEAT1_2; indicating that, if the experiment was performed using similarly efficient primers, NEAT1_1 is the primary isoform found in these samples. (i.e. in absolute molecule numbers: NEAT1 – NEAT1_2 = NEAT1_1) Please comment on the fact that in MM the levels of NEAT1_1 are strikingly increased. The fact that low NEAT1_2 levels correlate with a shorter time to first treatment is opposite for what is reported in the literature (i.e. high levels of NEAT1_2 confer poorer survival). Can the authors comment on this in the discussion? The title is vague and uninformative.

Author Response

  • Experimental methods, primers used, including efficiencies and ratio calculations are not reported. Choices of statistical tests are not justified (e.g. Rejection of Normal Distribution). See also below comment on pairwise comparisons.

We apologize for having omitted to report this information in the Methods section. Information concerning primers, evaluation of efficiency, ratio calculations, and statistical tests is given in the Appendix B of the revised version of the manuscript. The description of the methodological procedures has been amended accordingly.

Conclusions are either vague or largely overstated; i.e. in the abstract the authors claim that based on the data NEAT1 deserves further investigation on its possible specific role in CLL pathology, but most of the data presented in the manuscript shows no difference in expression or survival for CLL, and no independence to factors already known to be prognostic for patient outcomes. Similarly, the paper does not provide sufficient evidence to state that NEAT1_2 expression defines CLL subgroups.

Following the Reviewer’s criticism, we have modified these concepts in both abstract and discussion.

It is unclear why the authors chose to construct Kaplan Meiers of the top 75% of NEAT1 expressors with the bottom 25%. Could the authors construct the model with e.g. top 15% and bottom 15%? Does the separation occur in that case too, which would indicate it is because of NEAT1 expression, and not something else?

For convenience, here we report the response to a similar criticism from Reviewer #2. We apologize in that we have not clearly stated in the original version of the manuscript that p=0.038 should not be considered significant. The threshold for significance in the revised version has been retained at p=0.01.

For completeness, we analyzed the possible correlation between NEAT1 expression and TTFT either as a continuous variable or stratifying expression into n (2 up to 6) groups. Neither NEAT1 continuous expression nor any of the stratification correlated with outcome. The stratification into sextiles (close to the 15% suggested) was the only one that might indicate a trend of inferior TTFT associated with the lowest-expression sextile (if compared with all the other samples). However, such association was not absolutely retained in a multivariate model. 

For Reader’s convenience, we have reported these new data in the revised version and clearly stated in the text that no significant association between NEAT1_2 expression and prognosis could be highlighted in our dataset.

Minor issues:             

  • Line 62: cite appropriate references.

Done

  • Intro: introduce genetic architecture of NEAT1 gene (i.e. different isoforms and their proposed functions).

As suggested by the Reviewer, we added in the introduction (lane 57-71 of the revised version) some information on NEAT1 structure, isoforms and proposed functions, with appropriate references. Following this revision, in the revised version we deleted lines 99-101 that concerned NEAT1 isoforms description.

  • Line 71: please cite appropriate references for NEAT1 – p53 link.

Done

  • Line 79 : indicate abbreviation for MBL

Done

  • All qPCR results: using the 2^-dCT value on the Y-axis is less straightforward to interpret than using expression fold changes.

As requested by the Reviewer, all qPCR results have been reported as dCT in the revised version of the manuscript.

  • Line 108 – 110: Please report range, averages, and standard deviations for all samples.

As requested by the Reviewer, these data have been included in Figure 1c of the revised version of the manuscript.

  • Fig 1A: I am not sure about the author’s choice to compare their expression data in other tumors to those in CLL – it would make more sense to me to compare the data to the normal tissue counterparts if the authors would like to make a point about NEAT1’s importance for cancer.

We agree with the Reviewer that the comparison of each tumor type with its normal tissue counterpart would be of great interest. Unfortunately, we have not this possibility. However, it was not our intent to make a point about NEAT1’s importance for cancer, but rather to underline that the ratio NEAT1_2/NEAT1 in CLL is significantly higher than in all the other hematological tumors investigated.

  • Fig 2A: see comment above, the pairwise comparisons do not make sense to me – what is the biological rationale for comparing with 12+ vs the others? Is this the right control group?

As NEAT1_2 expression in CLL is quite heterogeneous and a proportion of samples showed high expression levels, our aim was to investigate whether such differences in NEAT1_2 expression could correlate with other characteristics, which usually define different CLL prognostic groups. As concerned cytogenetic alterations, we compared each prognostic group with the group without FISH abnormalities (none) that can be considered the control group in this analysis. The Kruskal Wallis one-way analysis of variance on ranks (KW) is the non-parametric method used to test whether the samples included in FISH groups originate from the same distribution. Dunn’s test was used to analyze all the specific group pairs for stochastic dominance in case of significant results from KW test. In the revised version, we have specified in the Figure legend that the p-values are referred to Dunn’s test results.

  • Please comment on the consistent higher levels of total NEAT1 versus NEAT1_2; indicating that, if the experiment was performed using similarly efficient primers, NEAT1_1 is the primary isoform found in these samples. (i.e. in absolute molecule numbers: NEAT1 – NEAT1_2 = NEAT1_1)

A comment on this has been included in the text, line 64-68.

  • Please comment on the fact that in MM the levels of NEAT1_1 are strikingly increased.

A comment on this has been included in the text, line 78-79.

  • The fact that low NEAT1_2 levels correlate with a shorter time to first treatment is opposite for what is reported in the literature (i.e. high levels of NEAT1_2 confer poorer survival). Can the authors comment on this in the discussion?

As concerns this aspect, please refer to previous point on the lack of correlation with TTFT, which prevents us to draw hypothesis.

  • The title is vague and uninformative.

The title of the revised version has been modified in: “NEAT1 long isoform is highly expressed in chronic lymphocytic leukemia irrespectively of cytogenetic groups or clinical outcome”

Round 2

Reviewer 3 Report

Figure 1 lacks Y axis labels.

The authors addressed all my concerns.